# Economic precarity, loneliness, and suicidal ideation during the COVID-19 pandemic

**Julia Raifman**[1]*, **Catherine K. Ettman**[2], **Lorraine T. Dean**[2], **Salma M. Abdalla**[1],
**Alexandra Skinner**[3], **Colleen L. Barry**[4], **Sandro Galea**[1]

1 Boston University School of Public Health, Boston, MA, United States of America, 2 Johns Hopkins University Bloomberg School of Public Health, Baltimore, MD, United States of America, 3 Brown University School of Public Health, Providence, RI, United States of America, 4 Cornell Jeb E. Brooks School of Public Policy, Ithaca, NY, United States of America

* jraifman@bu.edu

**Data Availability Statement:** The CLIMB dataset is publicly available here: https://dataverse.harvard.

## Abstract

The US population faced stressors associated with suicide brought on by the COVID-19 pandemic. Understanding the relationship between stressors and suicidal ideation in the context of the pandemic may inform policies and programs to prevent suicidality and suicide. We compared suicidal ideation between two cross-sectional, nationally representative surveys of adults in the United States: the 2017–2018 National Health and Nutrition Examination Survey (NHANES) and the 2020 COVID-19 and Life Stressors Impact on Mental Health and Well-being (CLIMB) study (conducted March 31 to April 13). We estimated the association between stressors and suicidal ideation in bivariable and multivariable Poisson regression models with robust variance to generate unadjusted and adjusted prevalence ratios (PR and aPR). Suicidal ideation increased from 3.4% in the 2017–2018 NHANES to 16.3% in the 2020 CLIMB survey, and from 5.8% to 26.4% among participants in low-income households. In the multivariable model, difficulty paying rent (aPR: 1.5, 95% CI: 1.2–2.1) and feeling alone (aPR: 1.9, 95% CI: 1.5–2.4) were associated with suicidal ideation but job loss was not (aPR: 0.9, 95% CI: 0.6 to 1.2). Suicidal ideation increased by 12.9 percentage points and was almost 4.8 times higher during the COVID-19 pandemic. Suicidal ideation was more prevalent among people facing difficulty paying rent (31.5%), job loss (24.1%), and loneliness (25.1%), with each stressor associated with suicidal ideation in bivariable models. Difficulty paying rent and loneliness were most associated with suicidal ideation. Policies and programs to support people experiencing economic precarity and loneliness may contribute to suicide prevention.

## Introduction

At the start of the coronavirus 2019 (COVID-19) pandemic, the population of the United States (US) faced several co-occurring stressors. The pandemic led to economic downturn, creating stressors of job loss and financial distress. Black, Hispanic, and Native American people are made especially vulnerable to these economic shocks as a result of centuries of structural

edu/dataset.xhtml?persistentId=doi:10.7910/DVN/
R2BWJL.

**Funding:** This study was funded in part through
support from the Rockefeller Foundation Boston
University 3-D Commission (https://
3dcommission.health). JR and AS worked on this
project while funded by K01 MH116817 from the
National Institutes of Health (https://www.nih.gov)
and grant No. 77922 from the Robert Wood
Johnson Foundation Evidence for Action Program
(https://www.rwjf.org). CE worked on this project
while funded by grant No. T32 AG 23482 15 from
the National Institutes of Health (https://www.nih.
gov). The funders had no role in study design, data
collection and analysis, decision to publish, or
preparation of the manuscript.

**Competing interests:** We have read the journal's
policy and the authors of this manuscript have the
following competing interests: SG reports serving
as a consultant for Sharecare and Tivity Health.
This does not alter our adherence to PLOS ONE
policies on sharing data and materials. All other
authors declare that no competing interests exist.

racism that shape inequities in wealth [1]. Additionally, physical distancing to prevent the
spread of COVID-19 created stressors including social isolation and loneliness.

Economic precarity [2] and social isolation [3] are associated with mental distress and sui-
cide. Other studies have documented increases in mental distress and suicidality [4–8] during
the pandemic, but few have examined economic hardship. While there were not increases in
suicide deaths in the US or many other countries in 2020 [9–11], suicide remains a leading
cause of premature death in the US [12] and may continue to evolve [13]. Suicidality is an indi-
cator of major mental distress and a risk factor for suicide [14]. Understanding the populations
most at risk of suicidal ideation and the association between COVID-19 stressors and suicidal
ideation can inform policies and programs to reduce suicidality and prevent suicide. This
study aimed to evaluate the relationship between stressors and suicidal ideation during the
start of the COVID-19 pandemic.

## Methods

### Sample

We used data from a nationally representative sample of US adults aged 18 or older collected
through the AmeriSpeak standing panel. Panelists were invited to participate in the COVID-
19 and Life Stressors Impact on Mental Health and Well-being (CLIMB) study from March
31, 2020 through April 13, 2020 and were paid a cash equivalent of $3 for completing the sur-
vey (64.3% completion rate). We created and applied post-stratification weights to align the
study sample with the U.S. adult population according to the U.S. Current Population Survey
[15]. Previously published work further describes details on the AmeriSpeak sampling frame
and the CLIMB study [5,16]. As a pre-pandemic comparison, we used data from the 2017–
2018 National Health and Nutrition Examination Survey (NHANES), a nationally representa-
tive sample of noninstitutionalized civilian US adults aged 18 years or older collected by the
US government. The CLIMB and NHANES samples are comparable in that both are nationally
representative groups of US adults residing in all 50 states and the District of Columbia. The
AmeriSpeak sampling frame covers approximately 97% of all US households, and NHANES
sampling units similarly cover all US counties. We excluded participants who did not respond
to questions about suicidal ideation in NHANES and participants who did not respond to any
analysis variables in CLIMB data.

### Exposures

We evaluated three COVID-19 stressors reflecting economic precarity and loneliness, each
measured as binary variables reported in response to a question, "Have any of the following
affected your life as a result of the coronavirus or COVID-19 outbreak?" Our exposure vari-
ables were based on whether participants checked boxes for "losing a job," "having difficulty
paying rent," and "feeling alone".

### Outcome

Both surveys assessed suicidal ideation based on Patient Health Questionnaire-9 (PHQ-9)
item 9, which asks participants whether they have had "Thoughts that you would be better off
dead or of hurting yourself in some way" over the past two weeks. Response options are "Not
at all," "several days," "more than half the days," or "nearly every day". We created a binary
variable for reporting these feelings with any frequency. Prior research indicates responses to
this question were correlated with future suicide attempts and deaths [14,17,18].

## Analysis

First, we described the demographic characteristics of participants in the 2020 CLIMB data and in the 2017–2018 NHANES data. Second, we estimated the prevalence of suicidal ideation by demographic characteristics and calculated the share with suicidal ideation within sub-groups in 2020 relative to 2017–2018. Third, we used CLIMB data to estimate unadjusted and adjusted prevalence ratios (PR and aPR) of the association between COVID-19 related stressors and suicidal ideation using a Poisson regression model with robust variance to approximate a log-binomial regression model, with $\alpha = 0.05$ [19]. In the multivariable model, to reduce bias, we adjusted for age group, education level, sex, race and ethnicity, household income, savings, marital status, COVID-19 illness, and COVID-19 bereavement. We ran a multivariate sensitivity analysis without the stressor variable for "having difficulty paying rent" in the same regression model as the "losing a job" exposure because difficulty paying rent may lie on the causal pathway in the association between job loss and suicidal ideation.

## Results

A total of 1,415 (96.3%) of 1,470 CLIMB participants responded to all questions relevant to the analysis and 5,085 (86.8%) of 5,856 NHANES participants responded to suicidal ideation questions and were included in the samples (Table 1). The NHANES sample was younger, less likely to be married, more likely to have high school or less education, and less likely to be low-income relative to the CLIMB sample.

**Table 1. Participant demographic characteristics.**

| | Sample characteristics | | | | |
| --- | --- | --- | --- | --- | --- |
| | NHANES, 2017–2018 | | CLIMB, 2020 | | p-value |
| | n | % | n | % | |
| **Total** | **5,085** | **100** | **1,415** | **100** | **N/A** |
| **Difficulty paying rent** | | | | | |
| No | N/A | N/A | 1,199 | 84.7 | N/A |
| Yes | N/A | N/A | 216 | 15.3 | N/A |
| **Lost job** | | | | | |
| No | N/A | N/A | 1,253 | 88.5 | N/A |
| Yes | N/A | N/A | 162 | 11.5 | N/A |
| **Feeling alone** | | | | | |
| No | N/A | N/A | 952 | 67.3 | N/A |
| Yes | N/A | N/A | 463 | 32.7 | N/A |
| Mean age (avg, std) | 49.6 | 18.5 | 46.0 | 16.5 | N/A |
| Age group | | | | | |
| 18–29 | 965 | 21.3 | 238 | 16.8 | <0.001 |
| 30–44 | 1,106 | 24.2 | 493 | 34.8 | <0.001 |
| 45–59 | 1,183 | 26.5 | 337 | 23.8 | 0,040 |
| 60+ | 1,831 | 28.1 | 347 | 24.5 | <0.001 |
| Sex | | | | | |
| Male | 2,489 | 48.7 | 708 | 50.0 | 0.387 |
| Female | 2,596 | 51.3 | 707 | 50.0 | 0.387 |
| Race/ethnicity | | | | | |
| Non-Hispanic White | 1,801 | 63.0 | 922 | 65.2 | 0.129 |
| Non-Hispanic Black | 1,178 | 11.1 | 137 | 9.7 | 0.133 |

*(Continued)*

**Table 1.** (Continued)

| | Sample characteristics | | | | |
|---|---|---|---|---|---|
| | NHANES, 2017–2018 | | CLIMB, 2020 | | p-value |
| | n | % | n | % | |
| **Total** | **5,085** | **100** | **1,415** | **100** | **N/A** |
| Non-Hispanic another race or multiracial | 947 | 10.1 | 105 | 7.4 | 0.002 |
| Hispanic | 1,159 | 15.8 | 251 | 17.7 | 0.086 |
| Education level | | | | | |
| High school graduate or less | 2,262 | 39.4 | 327 | 23.1 | <0.001 |
| Some college | 1,640 | 30.4 | 631 | 44.5 | <0.001 |
| College grad or more | 1,177 | 30.2 | 457 | 32.3 | 0.130 |
| Not reported | 6 | <0.1 | 0 | N/A | N/A |
| Marital status | | | | | |
| Unmarried or separated | 2,424 | 45.1 | 716 | 50.6 | <0.001 |
| Married | 2,412 | 51.5 | 699 | 49.4 | 0.162 |
| Not reported | 249 | 3.4 | 0 | N/A | <0.001 |
| Household income | | | | | |
| $0-$19,999 | 875 | 11.5 | 220 | 15.6 | <0.001 |
| $20,000-$39,999[a] | 1,322 | 21.4 | 321 | 22.7 | 0.294 |
| $40,000-$74,999[a] | 891 | 17.7 | 410 | 29.0 | <0.001 |
| ≥$75,000 | 1,356 | 38.7 | 464 | 32.8 | <0.001 |
| Not reported | 641 | 10.7 | 0 | N/A | <0.001 |
| Savings | | | | | |
| <$5,000 | N/A | N/A | 720 | 50.9 | N/A |
| ≥$5,000 | N/A | N/A | 695 | 49.1 | N/A |
| COVID-19 illness | | | | | |
| No | N/A | N/A | 1,403 | 99.1 | N/A |
| Yes | N/A | N/A | 12 | 0.9 | N/A |
| Death of someone close due to COVID-19 | | | | | , |
| No | N/A | N/A | 1,390 | 98.2 | N/A |
| Yes | N/A | N/A | 25 | 1.8 | N/A |

Notes: Percents are weighted. Data on suicidal ideation among those with COVID-19 illness and bereavement are based on small sample sizes.

[a] Income categories for NHANES participants were $20,000 to $44,999 and $45,000 to $74,999 based on different cut points for income questions.

The prevalence of suicidal ideation was 3.4% in the 2017–2018 NHANES sample and 16.3% in the 2020 CLIMB survey (Table 2). The greatest absolute increases in suicidal ideation and 2020 prevalence of suicidal ideation were among participants earning less than $20,000 (5.8% to 26.4%), participants aged 18 to 29 (4.1% to 23.5%), and participants who were Hispanic (3.7% to 23.1%). In 2020, suicidal ideation was high among those who faced difficulty paying rent (31.5%, Fig 1) and job loss (24.1%), as well as loneliness (25.1%).

Each of the stressors we evaluated were associated with suicidal ideation in the bivariable model (Table 3; Difficulty paying rent PR: 2.3, 95% CI: 1.8 to 3.1; feeling alone PR: 2.1, 95% CI: 1.6 to 2.6; job loss PR: 1.6, 95% CI: 1.1 to 2.2). In the multivariable model, difficulty paying rent was associated with suicidal ideation (aPR: 1.5, 95% CI: 1.2 to 2.1), while losing a job was not (aPR: 0.9, 95% CI: 0.6 to 1.2). The sensitivity analysis with job loss as the exposure without difficulty paying rent in the model did not change the effect estimate. Feeling alone was also associated with suicidal ideation (aPR: 1.9, 95% CI: 1.5 to 2.4). Although the sample size for persons with COVID-19 illness (n = 12) or bereavement (n = 25) is small, the results (66.7%

**Table 2. COVID-19 stressors, demographic characteristics, and suicidality.**

| | Suicidal ideation | | | | Absolute difference 2020 - (2017–2018) | Ratio, 2020: 2017–2018 | Z score p-value |
|---|---|---|---|---|---|---|---|
| | NHANES, 2017–2018, Suicidal ideation | | CLIMB, 2020 | | | | |
| | n | % | n | % | Percentage points | | |
| Total | 192 | 3.4 | 231 | 16.3 | 12.9 | 4.8 | <0.001 |
| **Difficulty paying rent** | | | | | | | |
| No | N/A | N/A | 163 | 13.6 | N/A | N/A | N/A |
| Yes | N/A | N/A | 68 | 31.5 | N/A | N/A | N/A |
| **Lost job** | | | | | | | |
| No | N/A | N/A | 192 | 15.3 | N/A | N/A | N/A |
| Yes | N/A | N/A | 39 | 24.1 | N/A | N/A | N/A |
| **Feeling alone** | | | | | | | |
| No | N/A | N/A | 115 | 12.1 | N/A | N/A | N/A |
| Yes | N/A | N/A | 116 | 25.1 | N/A | N/A | N/A |
| Age group | | | | | | | |
| 18–29 | 38 | 4.1 | 56 | 23.5 | 19.1 | 5.7 | <0.001 |
| 30–44 | 38 | 3.1 | 92 | 18.7 | 15.4 | 6.0 | <0.001 |
| 45–59 | 48 | 3.0 | 47 | 14.0 | 11 | 4.7 | <0.001 |
| 60+ | 68 | 3.5 | 36 | 10.4 | 6.8 | 2.9 | <0.001 |
| Sex | | | | | | | |
| Male | 102 | 4.0 | 109 | 15.4 | 11.5 | 3.9 | <0.001 |
| Female | 90 | 2.9 | 122 | 17.3 | 13.9 | 5.8 | <0.001 |
| Race/ethnicity | | | | | | | |
| Non-Hispanic White | 71 | 3.3 | 124 | 13.5 | 10.2 | 4.1 | <0.001 |
| Non-Hispanic Black | 33 | 3.1 | 22 | 16.1 | 12.1 | 4.9 | <0.001 |
| Non-Hispanic another race or multiracial | 34 | 4.0 | 27 | 25.7 | 20.1 | 6.0 | <0.001 |
| Hispanic | 54 | 3.7 | 58 | 23.1 | 19.1 | 6.2 | <0.001 |
| Education level | | | | | | | |
| High school graduate or less | 112 | 4.2 | 68 | 20.8 | 16.1 | 4.8 | <0.001 |
| Some college | 60 | 4.3 | 101 | 16 | 11.8 | 3.7 | <0.001 |
| College grad or more | 20 | 1.4 | 62 | 13.6 | 11.7 | 9.4 | <0.001 |
| Not reported | N/A | N/A | N/A | N/A | N/A | N/A | |
| Marital status | | | | | | | |
| Unmarried or separated | 127 | 4.6 | 152 | 21.2 | 16.4 | 4.6 | <0.001 |
| Married | 54 | 2.2 | 79 | 11.3 | 8.9 | 5.0 | <0.001 |
| Not reported | 11 | 5.2 | N/A | N/A | N/A | N/A | N/A |
| Household income | | | | | | | |
| $0-$19,999 | 50 | 5.8 | 58 | 26.4 | 19.8 | 4.4 | <0.001 |
| $20,000-$39,999[a] | 63 | 5.0 | 56 | 17.5 | 12.1 | 3.4 | <0.001 |
| $40,000-$74,999[a] | 32 | 3.3 | 59 | 14.4 | 11.1 | 4.4 | <0.001 |
| ≥$75,000 | 25 | 2.1 | 58 | 12.5 | 10.5 | 6.0 | <0.001 |
| Not reported | 22 | 2.7 | N/A | N/A | N/A | N/A | N/A |
| Savings | | | | | | | |
| <$5,000 | N/A | N/A | 155 | 21.5 | N/A | N/A | N/A |
| ≥$5,000 | N/A | N/A | 76 | 10.9 | N/A | N/A | N/A |
| COVID-19 illness | | | | | | | |
| No | N/A | N/A | 223 | 15.9 | N/A | N/A | N/A |

*(Continued)*

**Table 2.** (Continued)

| | Suicidal ideation | | | | Absolute difference 2020 - (2017–2018) | Ratio, 2020: 2017–2018 | Z score p-value |
|---|---|---|---|---|---|---|---|
| | NHANES, 2017–2018, Suicidal ideation | | CLIMB, 2020 | | | | |
| | n | % | n | % | Percentage points | | |
| Yes | N/A | N/A | 8 | 66.7 | N/A | N/A | N/A |
| Death of someone close due to COVID-19 | | | | | | | |
| No | N/A | N/A | 222 | 16 | N/A | N/A | N/A |
| Yes | N/A | N/A | 9 | 36 | N/A | N/A | N/A |

Notes: Percents are weighted. Data on suicidal ideation among those with COVID-19 illness and bereavement are based on small sample sizes. P-values depict Z-score differences for two sample tests of proportions.

[a] Income categories for NHANES participants were $20,000 to $44,999 and $45,000 to $74,999 based on different cut points for income questions.

and 36.0%, respectively) are suggestive that COVID-19 illness or bereavement may be associated with increased suicidal ideation.

## Discussion

We found that suicidal ideation was 4.8 times higher during the COVID-19 pandemic; 16.1% of people reported suicidal ideation in 2020, relative to 3.4% in 2017–2018. In keeping with

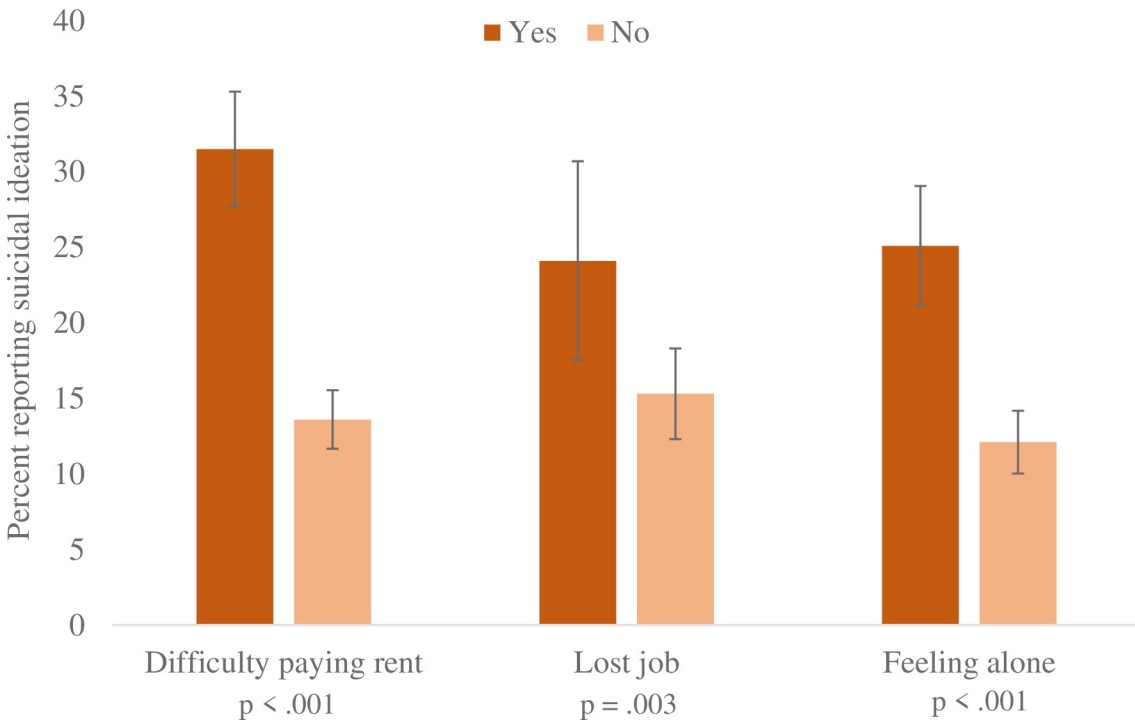

**Fig 1. COVID-19 stressors and suicidal ideation.** The figure depicts the percentages with 95% confidence intervals of participants who reported suicidal ideation among those who did or did not report COVID-19-related stressors, including difficulty paying rent, losing a job, and feeling alone. Data used to create the figure comes from the COVID-19 and Life Stressors Impact on Mental Health and Well-being (CLIMB) study conducted from March 31, 2020 through April 13, 2020.

**Table 3. COVID-19 related stressors and suicidal ideation over the past two weeks (N = 1415).**

| Variables | PR | 95% CI | p-value | aPR | 95% CI | p-value |
|---|---|---|---|---|---|---|
| **COVID-19 related stressors** | | | | | | |
| Lost job | 1.6 | 1.1–2.2 | 0.004 | 0.9 | 0.6–1.2 | 0.525 |
| Difficulty paying rent | 2.3 | 1.8–3.1 | <0.001 | 1.5 | 1.2–2.1 | 0.003 |
| Feeling alone | 2.1 | 1.6–2.6 | <0.001 | 1.9 | 1.5–2.4 | <0.001 |
| **Age group** | | | | | | |
| 18 to 29 | 2.3 | 1.5–3.3 | <0.001 | 1.3 | 0.9–2.0 | 0.217 |
| 30 to 44 | 1.8 | 1.3–2.6 | 0.001 | 1.3 | 0.9–1.9 | 0.147 |
| 45 to 59 | 1.3 | 0.9–2.0 | 0.155 | 1.1 | 0.7–1.7 | 0.627 |
| ≥60 | | Reference group | | | Reference group | |
| **Sex** | | | | | | |
| Male | | Reference group | | | Reference group | |
| Female | 1.1 | 0.9–1.4 | 0.345 | 0.9 | 0.7–1.1 | 0.414 |
| **Race/ethnicity** | | | | | | |
| Non-Hispanic White | | Reference group | | | Reference group | |
| Non-Hispanic Black | 1.2 | 0.8–1.8 | 0.404 | 0.8 | 0.5–1.2 | 0.288 |
| Non-Hispanic and another race or multiracial | 1.9 | 1.3–2.8 | 0.001 | 1.8 | 1.2–2.6 | 0.002 |
| Hispanic | 1.7 | 1.3–2.3 | <0.001 | 1.4 | 1.0–1.8 | 0.032 |
| **Income group** | | | | | | |
| $0-$19,999 | 2.1 | 1.5–2.9 | <0.001 | 1.2 | 0.8–1.8 | 0.317 |
| $20,000-$39,999 | 1.4 | 1.0–2.0 | 0.054 | 1.0 | 0.7–1.4 | 0.859 |
| $40,000-$74,999 | 1.2 | 0.8–1.6 | 0.414 | 1.0 | 0.7–1.4 | 0.872 |
| ≥$75,000 | | Reference group | | | Reference group | |
| **Education level** | | | | | | |
| High school graduate or less | 1.5 | 1.1–2.1 | 0.008 | 1.2 | 0.9–1.7 | 0.271 |
| Some college | 1.2 | 0.9–1.6 | 0.268 | 1.0 | 0.7–1.4 | 0.965 |
| College graduate or above | | Reference group | | | Reference group | |
| **Marital status** | | | | | | |
| Unmarried or separated | 1.9 | 1.5–2.4 | <0.001 | 1.3 | 1.0–1.8 | 0.036 |
| Married | | Reference group | | | Reference group | |
| **Savings** | | | | | | |
| <$5,000 | 2.0 | 1.5–2.5 | <0.001 | 1.4 | 1.1–1.9 | 0.021 |
| ≥$5,000 | | Reference group | | | Reference group | |
| **COVID-19 illness** | | | | | | |
| No | | Reference group | | | Reference group | |
| Yes | 4.2 | 2.8–6.4 | | 3.5 | 1.9–6.4 | <0.001 |
| **Death of someone close due to COVID-19** | | | | | | |
| No | | Reference group | | | Reference group | |
| Yes | 2.3 | 1.3–3.9 | | 1.7 | 0.9–3.0 | 0.109 |

Notes: Estimates are prevalence ratios (PR) and adjusted prevalence ratios (aPR) based on Poisson regression analyses with robust variance. The PR is based on a bivariable analyses of each variable and suicidal ideation. The aPR is adjusted for all variables listed in the table. There were small samples of participants with COVID-19 illness (n = 12) and bereavement (n = 25).

prior studies, we found that people living in low-income households are particularly at risk of mental distress during the COVID-19 pandemic [5,6,16].

Reporting difficulty paying rent was associated with suicidal ideation. Prior research indicates that financial distress, such as that which became widespread during the COVID-19 pandemic, is associated with suicide[2] and that eviction in particular is associated with suicide [20].

Policies such as the Centers for Disease Control and Prevention's federal eviction moratorium, state eviction moratoriums, and federal and state unemployment insurance policies [21] and federal stimulus payments may help prevent suicide. While job loss was not associated with suicidal ideation during the CLIMB study period of late March and early April in the adjusted analysis, it is important to study job loss and mental health over a longer term period as high unemployment was prolonged for several months, especially for people in low-income households and who are Black and Hispanic [22]. It is also possible that the suicidality impacts of job loss differed by wealth and whether people who lost work faced imminent economic hardship, and further studies among subgroups are needed.

People who reported feeling alone were 1.9 times as likely to report suicidal ideation, highlighting the need for programs and policies to provide social support, such as through social connections in environments with lower COVID-19 risk (e.g. outdoors) or via computer or phone. There is a need for further research on loneliness among subgroups such as older populations.

This study was conducted early in the course of COVID-19 spread across the US; we did not have a large enough sample of persons experiencing COVID-19 illness or bereavement to study the association between COVID-19 illness or bereavement with suicidal ideation. The results suggest a potential association between these exposures and suicidal ideation that warrants further study.

Finally, prior studies indicate that means restriction [23], particularly of firearms [24], is associated with reductions in suicide. Suicide by firearm is the predominant means of suicide death in the United States. As such, and in light of previous evidence that links suicidality to suicide death, policies or programs to reduce household firearm ownership could play an important role in suicide prevention in the COVID-19 context of elevated stressors.

Limitations include that the characteristics of participants in CLIMB and NHANES differed, although both were nationally representative surveys and should be generalizable to the broader population. The NHANES sample was younger and less likely to be low-income. We were further limited by the questions asked in the surveys as indicators of economic precarity and loneliness and the possibility that participants interpreted these questions in different ways. These questions were not included in the NHANES survey, so it was not possible to compare these exposures before and after the pandemic. Furthermore, the CLIMB study was conducted early in the pandemic, and the relationship between stressors and suicidal ideation may have changed as the pandemic and associated stressors continue to affect the US population. There is a need for further research on exposures and suicidality over time and among subgroups, especially those affected by structural racism and inequities.

## Conclusion

Suicidal ideation increased substantially during the COVID-19 pandemic. Those facing difficulty paying rent and feeling alone may be at particular risk of suicide. Policies and programs to support people experiencing economic precarity and difficulty paying rent may contribute to suicide prevention, as may programs to support individuals facing prolonged social isolation.

## Supporting information

**S1 Checklist. STROBE statement—checklist of items that should be included in reports of *cross-sectional studies*.**
(DOCX)

## Author Contributions

**Conceptualization:** Julia Raifman, Catherine K. Ettman, Lorraine T. Dean, Salma M. Abdalla, Alexandra Skinner, Colleen L. Barry, Sandro Galea.

**Data curation:** Julia Raifman, Catherine K. Ettman.

**Formal analysis:** Julia Raifman, Colleen L. Barry.

**Funding acquisition:** Catherine K. Ettman, Salma M. Abdalla, Sandro Galea.

**Investigation:** Lorraine T. Dean.

**Methodology:** Julia Raifman, Catherine K. Ettman, Lorraine T. Dean, Salma M. Abdalla, Alexandra Skinner, Colleen L. Barry, Sandro Galea.

**Project administration:** Julia Raifman, Salma M. Abdalla, Alexandra Skinner.

**Visualization:** Julia Raifman, Lorraine T. Dean, Salma M. Abdalla, Alexandra Skinner, Colleen L. Barry, Sandro Galea.

**Writing – original draft:** Julia Raifman.

**Writing – review & editing:** Catherine K. Ettman, Lorraine T. Dean, Salma M. Abdalla, Alexandra Skinner, Colleen L. Barry, Sandro Galea.

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
