## [Decision Letter · Decision Letter 0]

2 Nov 2021

PONE-D-21-30553Economic precarity, social isolation, and suicidal ideation during the COVID-19 pandemicPLOS ONE

Dear Dr. Raifman,

Thank you for submitting your manuscript to PLOS ONE. After careful consideration, we feel that it has merit but does not fully meet PLOS ONE’s publication criteria as it currently stands. Therefore, we invite you to submit a revised version of the manuscript that addresses the points raised during the review process. There is consensus across the Reviewer's that this manuscript would make a welcome contribution to the literature on the impacts of COVID-19, however, there are concerns as to how well the literature reviewed as part of the Introduction aligns to the aims and outcomes of this study, as well as insufficient detail in the Methodology. Many of the issues raised are quite minor, however, I strongly encourage you to address all feedback, point-by-point, to strengthen the manuscript.  In particular, please make sure to complete and attach the STROBE statement as part of your re-submission, as per Reviewer 1's request.

We look forward to receiving your revised manuscript.

Kind regards,

Michelle Torok, Ph.D.

Academic Editor

PLOS ONE

Journal Requirements:

[We have read the journal's policy and the authors of this manuscript have the following competing interests: SG reports serving as a consultant for Sharecare and Tivity Health. All other authors declare that no competing interests exist.]

Additional Editor Comments:

In addition to the comments provided by Reviewer's 1 and 2, the following minor issues need to be addressed:

1. The authors should revise the Introduction to focus on the suicidal ideation literature rather than the mortality literature, consistent with the outcomes measured in their study.

2. In the exposures section (Methods) the authors state that social isolation is measured by the question "feeling alone" - yet refer to to this exposure as loneliness in the Results. Loneliness is more accurate given that feeling alone does not capture whether the person is actually alone. Please change social isolation to loneliness in the Methods.

3. This paper would benefit from more description of where the two samples were drawn from (geographically), the recruitment methods (were similar methods used?), and the response rates for each survey to help the reader understand how comparable the two samples are. Could authors please also include the mean age for the two samples in Table 1, as it looks like the CLIMB sample could be overall younger than the NHANES sample.

4. In the results, the authors make a statement about the increase in suicidal ideation between the two surveys ["Overall, suicidal ideation increased more

107 than fourfold, from 3.4% in the 2017-2018 NHANES to 16.3% in the 2020 CLIMB survey"] however I'm not sure it's appropriate to use 'increase' here given that two samples are not from the same population (and we do not know if the CLIMB sample would have had a higher prevalence of ideation outside of COVID]; re-writing as something more objective would be beneficial, like "the prevalence of suicidal ideation was 3.4% in the NHANES study and 16.3% in the CLIMB sample".

Reviewers' comments:

Reviewer's Responses to Questions

**Comments to the Author**

1. Is the manuscript technically sound, and do the data support the conclusions?

Reviewer #1: Yes

Reviewer #2: No

2. Has the statistical analysis been performed appropriately and rigorously? 

Reviewer #1: Yes

Reviewer #2: I Don't Know

3. Have the authors made all data underlying the findings in their manuscript fully available?

Reviewer #1: Yes

Reviewer #2: Yes

4. Is the manuscript presented in an intelligible fashion and written in standard English?

Reviewer #1: Yes

Reviewer #2: Yes

5. Review Comments to the Author

Reviewer #1: I thank the authors for the interesting and important study that has further novelty and importance by stratifying analyses by key demographic groups likely disproportionately affected by the pandemic. The authors’ findings confirm this disproportionate impact is the case. My comments are all minor I think, with the most major probably pertaining to ensuring the manuscript is reported per the STROBE Statement and the RECORD Statement (if it’s routinely collected data - NHANES is arguably but unsure about AmeriSpeak) and these reporting guidelines are populated and uploaded.

Page 2, line 30: ‘analyzed April 28 to September 30, 2020’ – the authors may not need to report the dates they analysed the CLIMB data in the abstract, unless this is in here for some specific reason?

Page 2, line 39 ‘Suicidal ideation increased more than fourfold during the COVID-19 pandemic.’ – could say ‘increased by 12.9% and was almost 4.8 times higher.’ It’s definitely closer to five than four.

Page 2, line 41 ‘support people experiencing economic precarity’ – Although ‘precarity’ is a simpler version of precarious and precariousness I imagine, I hard not heard of it before. Is ‘uncertainty’ a more common word in your context, or ‘difficulties’ or ‘challenges’ even? It’s not a big issue so I will leave it to the authors to decide.

Page 3, line 55: ‘Increases in suicide have occurred with prior pandemics [4]’ – Here, it would be best to instead cite these two reviews:

Rogers, J. P., Chesney, E., Oliver, D., Begum, N., Saini, A., Wang, S., ... & David, A. S. (2021). Suicide, self-harm and thoughts of suicide or self-harm in infectious disease epidemics: a systematic review and meta-analysis. Epidemiology and psychiatric sciences, 30.

Zortea, T. C., Brenna, C. T., Joyce, M., McClelland, H., Tippett, M., Tran, M. M., ... & Platt, S. (2020). The impact of infectious disease-related public health emergencies on suicide, suicidal behavior, and suicidal thoughts. Crisis.

Page 3, line 57: ‘While suicide mortality data are typically not publicly available for several months after the end of each year, understanding the populations most at risk of suicidal ideation and the association between COVID-19 stressors and suicidal ideation can inform policies and programs to prevent suicide.’ – I think you will need to change this statement to incorporate the following papers:

Pirkis, J., John, A., Shin, S., DelPozo-Banos, M., Arya, V., Analuisa-Aguilar, P., ... & Spittal, M. J. (2021). Suicide trends in the early months of the COVID-19 pandemic: an interrupted time-series analysis of preliminary data from 21 countries. The Lancet Psychiatry, 8(7), 579-588.

Knipe, D., John, A., Padmanathan, P., Eyles, E., Dekel, D., Higgins, J. P., ... & Gunnell, D. (2021). Suicide and self-harm in low-and middle-income countries during the COVID-19 pandemic: A systematic review. medRxiv.

Farooq, S., Tunmore, J., Ali, W., & Ayub, M. (2021). Suicide, self-harm and suicidal ideation during COVID-19: a

systematic review. Psychiatry Research, 114228.

John, A., Eyles, E., Webb, R. T., Okolie, C., Schmidt, L., Arensman, E., ... & Gunnell, D. (2021). The impact of the COVID-19 pandemic on self-harm and suicidal behaviour: update of living systematic review. F1000Research, 9(1097), 1097.

Although many of these papers are about suicide mortality data, I think this is perhaps less relevant for the authors, as most indices of mental health seem to have changed except suicide mortality data (I only know of increases in suicide mortality data in Vienna, Puerto Rico and Japan at the moment). Therefore, it’d probably be good to cite this in your introduction too:

Farooq, S., Tunmore, J., Ali, W., & Ayub, M. (2021). Suicide, self-harm and suicidal ideation during COVID-19: a systematic review. Psychiatry Research, 114228.

People will make arguments that those who think, attempt and die by suicide are distinct, and that these populations are distinct from people experiencing mental health symptoms or conditions too, so I think it’d be good to focus on the relevant literature regarding ideation specifically.

Page 3, line 66: ‘AmeriSpeak standing panel’ – can the authors provide some more information on this? While NHANES is well-known internationally, I have never heard of this. Doing a quick Google Search, some further elaboration would alleviate any reader concerns about the rigor of this survey. A recent paper in the Lancet Americas spent some more time describing it: https://www.thelancet.com/journals/lanam/article/PIIS2667-193X%2821%2900087-9/fulltext

You might even just paraphrase what they said, which is: ‘Details on the AmeriSpeak sampling frame and on the CLIMB study can be found in previous writing. [[1],[17]]’

Page 3, line 68: ‘(64% completed)’ – Do you mean a 64% response rate? Was the survey considered nationally representative before non-response? It seems paraphrasing a statement like this might help: ‘Post-stratification weights were created; once applied, the survey weights aligned the study sample with the U.S. adult population based on the U.S. Current Population Survey. [[19]]’ https://www.thelancet.com/journals/lanam/article/PIIS2667-193X%2821%2900087-9/fulltext Could also add the decimal place: 64.3

Page 3, line 73: We excluded participants who did not respond to questions about suicidal ideation in NHANES and participants who did not respond to any analysis variables in CLIMB data.’ – I presume the results section will mention how many people were excluded?

Page 4, line 77: ‘We evaluated three COVID-19 stressors reflecting economic precarity and social isolation, each measured as binary variables reported in response to a question, “Have any of the following affected your life as a result of the coronavirus or COVID-19 outbreak?” First, we measured job loss based on checking “losing a job.” Second, we measured difficulty paying rent based on checking “having difficulty paying rent.” Third, we measured social isolation as checking “feeling alone.”’

– I think the authors’ findings are really interesting and important as it is nice to see a focus on inequity with these outcomes. So, could the authors clarify that these were the only available stressors for economic precarity and social isolation, and if there were reasons for not looking at other aspects (e.g., outside the scope of the paper)?

Also, it may be odd to say ‘we measured’ if you did not select the measures or collect the data. You could say “The three checkboxes we used asked about ‘losing a job,’ ‘having difficulty paying rent,’ and “feeling alone.”

Page 4, line 85: ‘We measured suicidal ideation’ – could say that ‘both surveys assessed suicidal ideation.’

Page 4, line 89: ‘Prior research indicates responses to this question were correlated with future suicide attempts and deaths [7-9].’ – if they are future suicide attempts and deaths, do you need to say ‘predict’ instead of ‘were correlated with’, and say how well they predict, as any one risk factor by itself is likely a poor predictor of attempts and deaths – see https://pubmed.ncbi.nlm.nih.gov/27841450/

Page 4, line 96: It is great the authors have opted with PRs and Poisson rather than ORs and log reg. Is it easy for the authors to supply their code for the analysis in case other researchers may wish to repeat their study with similar data?

Page 5, line 106: ‘Overall, suicidal ideation increased more than fourfold, from 3.4% in the 2017-2018 NHANES to 16.3% in the 2020 CLIMB survey.’ – As it’s closer to fivefold the authors could say ‘4.79’ or ‘4.8,’ perhaps one decimal place is best.

Page 6, line 116 and page 8, line 147. Can the authors please have columns reporting precise rather than thresholded p-values? This will avoid Type I and Type II errors

(https://www.ncbi.nlm.nih.gov/pmc/articles/PMC2850991/), align with the ASA Statement on p-values (https://www.tandfonline.com/doi/full/10.1080/00031305.2016.1154108) and guidelines from other influential papers on p-values (https://www.ncbi.nlm.nih.gov/pmc/articles/PMC4877414/).

Page 9, line 156: ‘more than fourfold’ – 4.8 times higher? Or nearly fivefold.

Page 9, line 164: ‘Center for Disease Control and Prevention’ – ‘Centers’

Page 9, line 166: ‘may play help’ – typo?

Page 9, line 172: ‘People who reported feeling lonely were nearly twice as likely to report suicidal ideation’ – 1.9 times as likely? Also, there is some inconsistency in terminology here. The question AmeriSpeak asked was about feeling alone, which you’ve termed social isolation, but it’s referred to as ‘feeling lonely’ in the discussion. Maybe for consistency and brevity you can say ‘feeling alone’ throughout, as it depends on how people interpreted this question (were they actually socially isolated, or perceiving being socially isolated and feeling alone and lonely).

Page 10, line 167: ‘While job loss was not associated with suicidal ideation during the CLIMB study period of late March and early April,’ – do you need to suffix with ‘in the adjusted analysis’?

Page 10, line 183: ‘particularly of firearms [15], is associated with reductions in suicide.’

- A lot of the means restriction interventions were recently summarised in: Ishimo, M. C., Sampasa-Kanyinga, H., Olibris, B., Chawla, M., Berfeld, N., Prince, S. A., ... & Lang, J. J. (2021). Universal interventions for suicide prevention in high-income Organisation for Economic Co-operation and Development (OECD) member countries: a systematic review. Injury prevention, 27(2), 184-193. I suggest citing it.

For firearms specifically, I think the latest systematic review is this, but I suggest the authors citation search it: https://academic.oup.com/epirev/article/38/1/140/2754868?login=true Also better to cite though than the currently cited study.

You might also need to say here that firearms is the predominant suicide method in the US (as it might be hanging elsewhere) so that international readers know why firearms warrant mentioning here.

Page 10, line 188 ‘Limitations include that suicidal ideation was based on self-report’ – yes, any self-reported measure has limitations but how else do we measure ideation? If there is not viable alternative way to measure ideation, I suggest the authors omit this as it’s not something that can be improved upon and it’s a universal limitation of studies on ideation.

Page 10, line 188 ‘that the characteristics of participants in CLIMB and NHANES differed.’ – But both were nationally representative surveys? Additionally, you should probably note here that suicidal ideation was still higher in 2020 (I don’t mean ‘statistical significance,’ just higher) across all strata, compared to 2017 to 2018.

Page 10, line 189 ‘Those who responded to surveys may have differed from those who did not, particularly if stressors affected survey participation.’ – I’m not a survey expert, but as I understand, post-stratification weighting should cancel this out by adjusting for non-response?

Page 10, lines 188-193 – It would perhaps be good to include something about the limitations or exposure measurement if you think there are some. 1-item measures are sometimes criticised. ‘having difficulty paying rent’ is fairly straightforward, although it does not account for people’s income levels. ‘Losing a job’ could mean different things to different people – what about a contractor who has lots of small jobs and loses one job. ‘Feeling alone’ seems a little open to interpretation. These are only minor limitations, but may just warrant mentioning.

Page 10, line 186: Those facing difficulty paying rent and loneliness may be at particular risk of suicide.’ – maybe ‘feeling alone’ instead of ‘loneliness’ for consistency?

Page 14, Fig 1: I’m not sure the data labels add to the figure as they are already in the tables, so they could be removed. As they are above the upper limit, they might be misinterpreted as the upper limit.

Page 14, Fig 1: Red is sometimes considered a colour to avoid in the suicide sector due to the links to blood. The authors could instead choose from these colorblind-friendly colors: https://jfly.uni-koeln.de/color/#pallet

As this is an observational study, per PLOS One requirements the authors need to report according to, populate and submit the STROBE Statement. If the authors consider that NHANES and AmeriSpeak are routinely collected health data, the RECORD Statement (considered supplementary to STROBE and submitted alongside it) can be populated and submitted as well.

Overall, I think this is a really good manuscript. As the literature on suicide is moving so quickly during COVID-19 I’d encourage the authors to check the literature for new studies and reviews just prior to submission if invited to resubmit, be as outcome-specific as possible (ideation, not attempts and deaths), and emphasise slightly more the novelty of this paper (have many other papers done this?) focusing on inequities and people disproportionately bearing the impacts of COVID-19.

Reviewer #2: This is an interesting study on the increase in the prevalence of suicidal ideation during the COVID-19 pandemic.

However, I do not recommend this article for publication in its current form as several limitations have to be highlighted:

1. "The CLIMB and NHANES samples are comparable in that both are nationally representative": in my opinion, this statement is one of the major limitations of the study and should be more clearly analyzed and discussed. The Table 2 should try to depict more precisely the comparability of the two samples. Notably, the percentage of people with high school graduate or less is higher in the NHANES compared to the CLIMBS.

2. The prevalence of "feeling alone", "difficulty paying a rent" or "lost job" were not assessed in the NHANES study so that it is not possible to know if the prevalence of these social conditions were similar or not in the two samples. Moreover, economic precarity and loneliness are already identified as robust risk factors for suicidal ideation and behaviors so that this result does not add much value to the existing evidence regarding suicide prevention.

3. It is not clear if the prevalence ratios were measured with the data issued from the two samples or only for the CLIMB study? As some variables were measured in the two samples and other only in the CLIMB sample, the statistic analyses should be more precisely described and explained.

4. Figure 1: the p should be given for a better understanding of the results depicted in this figure. Are the observed differences statistically significant?

5. The discussion does not fit with the results of the study. For example, the authors discuss the role of means restrictions to prevent suicide, while their study is focused on suicidal ideation with no assessment of access to suicide means.

6. "We found that people living in low-income households and young people are particularly at risk of mental distress during the COVID-19 pandemic": this statement is not supported by the multivariate analysis showing no differences in reported suicidal ideation according to the age of participants

6. PLOS authors have the option to publish the peer review history of their article (what does this mean?). If published, this will include your full peer review and any attached files.

Reviewer #1: **Yes: **Stuart Leske

Reviewer #2: **Yes: **Edouard Leaune

---

## [Author Response · Author response to Decision Letter 0]

27 May 2022

Thank you for the opportunity to revise and respond to the reviewers’ comments on our manuscript, entitled “Economic precarity, social isolation, and suicidal ideation during the COVID-19 pandemic”. We appreciate the comments and the editor’s and reviewers’ thoughtful feedback that has improved the quality of this manuscript. We have provided detailed responses below and tracked changes to the manuscript. 

Journal Requirements:

The manuscript has been edited to fit specific journal requirements. 

[We have read the journal's policy and the authors of this manuscript have the following competing interests: SG reports serving as a consultant for Sharecare and Tivity Health. All other authors declare that no competing interests exist.]

We added the following Competing Interests statement to the cover letter: “We have read the journal’s policy and the authors of this manuscript have the following competing interests: SG reports serving as a consultant for Sharecare and Tivity Health. This does not alter our adherence to PLOS ONE policies on sharing data and materials. All other authors declare that no competing interests exist.”

Thank you for this clarification. There are no ethical or legal restrictions to sharing the CLIMB dataset, so we uploaded the minimal anonymized dataset as a Supporting Information file. The full dataset may be available upon reasonable request of the authors and will be considered on a case-by-case basis by CE and SG. NHANES is a publicly available dataset made available online by the CDC: https://wwwn.cdc.gov/nchs/nhanes/continuousnhanes/default.aspx?BeginYear=2017. We updated the cover letter to reflect these changes to our Data Availability statement.

Additional Editor Comments:

In addition to the comments provided by Reviewers 1 and 2, the following minor issues need to be addressed:

1. The authors should revise the Introduction to focus on the suicidal ideation literature rather than the mortality literature, consistent with the outcomes measured in their study.

Thank you for this feedback. We revised the introduction to focus more specifically on suicidal ideation with references to recent literature in the context of the COVID-19 pandemic. The introduction now includes the following text: “Economic precarity [2] and social isolation [3] are associated with mental distress and suicide. Other studies have documented increases in mental distress and suicidality [4-8] during the pandemic, but few have examined economic hardship. While there were not increases in suicide deaths in the US or many other countries in 2020 [9-11], suicide remains a leading cause of premature death in the US [12] and may continue to evolve [13]. Suicidality is an indicator of major mental distress and a risk factor for suicide [14]. Understanding the populations most at risk of suicidal ideation and the association between COVID-19 stressors and suicidal ideation can inform policies and programs to reduce suicidality and prevent suicide.”

2. In the exposures section (Methods) the authors state that social isolation is measured by the question "feeling alone" - yet refer to this exposure as loneliness in the Results. Loneliness is more accurate given that feeling alone does not capture whether the person is actually alone. Please change social isolation to loneliness in the Methods.

Thank you for this comment. We changed “social isolation” to “loneliness” in the Methods as suggested and in the title of the manuscript.

3. This paper would benefit from more description of where the two samples were drawn from (geographically), the recruitment methods (were similar methods used?), and the response rates for each survey to help the reader understand how comparable the two samples are. Could authors please also include the mean age for the two samples in Table 1, as it looks like the CLIMB sample could be overall younger than the NHANES sample.

Thank you for this suggestion. Based on your feedback and comments from Reviewer 1, we added the following sentences about the CLIMB sample: “We created and applied post-stratification weights to align the study sample with the U.S. adult population according to the U.S. Current Population Survey [15]. Previously published work further describes details on the AmeriSpeak sampling frame and the CLIMB study [5,16].” We also described that CLIMB has a 64.3% completion rate. To further describe the comparability between the two samples, we added the following sentences: “The CLIMB and NHANES samples are comparable in that both are nationally representative groups of US adults residing in all 50 states and the District of Columbia. The AmeriSpeak sampling frame covers approximately 97% of all US households, and NHANES sampling units similarly cover all US counties.” We revised Table 1 to include the mean age for the two samples, and we added the following sentence to the Results section to describe differences: “The NHANES sample was younger, less likely to be married, more likely to have high school or less education, and less likely to be low-income relative to the CLIMB sample.” 

4. In the results, the authors make a statement about the increase in suicidal ideation between the two surveys ["Overall, suicidal ideation increased more

107 than fourfold, from 3.4% in the 2017-2018 NHANES to 16.3% in the 2020 CLIMB survey"] however I'm not sure it's appropriate to use 'increase' here given that two samples are not from the same population (and we do not know if the CLIMB sample would have had a higher prevalence of ideation outside of COVID]; re-writing as something more objective would be beneficial, like "the prevalence of suicidal ideation was 3.4% in the NHANES study and 16.3% in the CLIMB sample". 

Thank you for this suggestion. We replaced the sentence you cited with the following: “The prevalence of suicidal ideation was 3.4% in the 2017-2018 NHANES sample and 16.3% in the 2020 CLIMB survey.”

Reviewer #1: I thank the authors for the interesting and important study that has further novelty and importance by stratifying analyses by key demographic groups likely disproportionately affected by the pandemic. The authors’ findings confirm this disproportionate impact is the case. My comments are all minor I think, with the most major probably pertaining to ensuring the manuscript is reported per the STROBE Statement and the RECORD Statement (if it’s routinely collected data - NHANES is arguably but unsure about AmeriSpeak) and these reporting guidelines are populated and uploaded.

R1 Comment 1

Page 2, line 30: ‘analyzed April 28 to September 30, 2020’ – the authors may not need to report the dates they analysed the CLIMB data in the abstract, unless this is in here for some specific reason?

R1 Response 1

Thank you for this comment. We deleted the following phrase from the abstract: “analyzed April 28 to September 30, 2020.”

R1 Comment 2

Page 2, line 39 ‘Suicidal ideation increased more than fourfold during the COVID-19 pandemic.’ – could say ‘increased by 12.9% and was almost 4.8 times higher.’ It’s definitely closer to five than four.

R1 Response 2

Thank you for this suggestion. We replaced “more than fourfold” with “increased by 12.9 percentage points and was almost 4.8 times higher”.

R1 Comment 3

Page 2, line 41 ‘support people experiencing economic precarity’ – Although ‘precarity’ is a simpler version of precarious and precariousness I imagine, I had not heard of it before. Is ‘uncertainty’ a more common word in your context, or ‘difficulties’ or ‘challenges’ even? It’s not a big issue so I will leave it to the authors to decide.

R1 Response 3

Thank you for this comment. We appreciate your perspective, but we think “precarity” best captures this experience. We have seen authors use the term previously in several contexts including the following: 

Perry BL, Aronson B, Pescosolido BA. Pandemic precarity: COVID-19 is exposing and exacerbating inequalities in the American Heartland. PNAS. 2021;118(8):e2020685118.

Bozarth K, Western G, Jones J. Black women best: The framework we need for an equitable economy. Roosevelt Institute. 2020. Available from: https://rooseveltinstitute.org/wp-content/uploads/2020/09/RI_Black-Women-Best_IssueBrief-202009.pdf.

R1 Comment 4

Page 3, line 55: ‘Increases in suicide have occurred with prior pandemics [4]’ – Here, it would be best to instead cite these two reviews:

Rogers, J. P., Chesney, E., Oliver, D., Begum, N., Saini, A., Wang, S., ... & David, A. S. (2021). Suicide, self-harm and thoughts of suicide or self-harm in infectious disease epidemics: a systematic review and meta-analysis. Epidemiology and psychiatric sciences, 30.

Zortea, T. C., Brenna, C. T., Joyce, M., McClelland, H., Tippett, M., Tran, M. M., ... & Platt, S. (2020). The impact of infectious disease-related public health emergencies on suicide, suicidal behavior, and suicidal thoughts. Crisis.

R1 Response 4

Thank you for sharing these reviews. We added citations to both articles.

R1 Comment 5

Page 3, line 57: ‘While suicide mortality data are typically not publicly available for several months after the end of each year, understanding the populations most at risk of suicidal ideation and the association between COVID-19 stressors and suicidal ideation can inform policies and programs to prevent suicide.’ – I think you will need to change this statement to incorporate the following papers:

Pirkis, J., John, A., Shin, S., DelPozo-Banos, M., Arya, V., Analuisa-Aguilar, P., ... & Spittal, M. J. (2021). Suicide trends in the early months of the COVID-19 pandemic: an interrupted time-series analysis of preliminary data from 21 countries. The Lancet Psychiatry, 8(7), 579-588.

Knipe, D., John, A., Padmanathan, P., Eyles, E., Dekel, D., Higgins, J. P., ... & Gunnell, D. (2021). Suicide and self-harm in low-and middle-income countries during the COVID-19 pandemic: A systematic review. medRxiv.

Farooq, S., Tunmore, J., Ali, W., & Ayub, M. (2021). Suicide, self-harm and suicidal ideation during COVID-19: a systematic review. Psychiatry Research, 114228.

John, A., Eyles, E., Webb, R. T., Okolie, C., Schmidt, L., Arensman, E., ... & Gunnell, D. (2021). The impact of the COVID-19 pandemic on self-harm and suicidal behaviour: update of living systematic review. F1000Research, 9(1097), 1097.

Although many of these papers are about suicide mortality data, I think this is perhaps less relevant for the authors, as most indices of mental health seem to have changed except suicide mortality data (I only know of increases in suicide mortality data in Vienna, Puerto Rico and Japan at the moment). Therefore, it’d probably be good to cite this in your introduction too:

Farooq, S., Tunmore, J., Ali, W., & Ayub, M. (2021). Suicide, self-harm and suicidal ideation during COVID-19: a systematic review. Psychiatry Research, 114228.

People will make arguments that those who think, attempt and die by suicide are distinct, and that these populations are distinct from people experiencing mental health symptoms or conditions too, so I think it’d be good to focus on the relevant literature regarding ideation specifically.

R1 Response 5

Thank you for these suggestions. We removed the phrase about suicide mortality data not typically being publicly available for several months after the end of each year. Based on your thoughts and comments from the editor, we also revised the introduction section to focus more specifically on suicidal ideation rather than suicide mortality. We greatly appreciate the recent literature you recommended and we cited these sources. The introduction now includes the following text: “Economic precarity [2] and social isolation [3] are associated with mental distress and suicide. Other studies have documented increases in mental distress and suicidality [4-8] during the pandemic, but few have examined economic hardship. While there were not increases in suicide deaths in the US or many other countries in 2020 [9-11], suicide remains a leading cause of premature death in the US [12] and may continue to evolve [13]. Suicidality is an indicator of major mental distress and a risk factor for suicide [14]. Understanding the populations most at risk of suicidal ideation and the association between COVID-19 stressors and suicidal ideation can inform policies and programs to reduce suicidality and prevent suicide.”

R1 Comment 6

Page 3, line 66: ‘AmeriSpeak standing panel’ – can the authors provide some more information on this? While NHANES is well-known internationally, I have never heard of this. Doing a quick Google Search, some further elaboration would alleviate any reader concerns about the rigor of this survey. A recent paper in the Lancet Americas spent some more time describing it: https://www.thelancet.com/journals/lanam/article/PIIS2667-193X%2821%2900087-9/fulltext

You might even just paraphrase what they said, which is: ‘Details on the AmeriSpeak sampling frame and on the CLIMB study can be found in previous writing. [[1],[17]]’

R1 Response 6

Thank you for this suggestion. We cited these papers and added a clarifying sentence: “Previously published work further describes details on the AmeriSpeak sampling frame and the CLIMB study [5,16].”

R1 Comment 7

Page 3, line 68: ‘(64% completed)’ – Do you mean a 64% response rate? Was the survey considered nationally representative before non-response? It seems paraphrasing a statement like this might help: ‘Post-stratification weights were created; once applied, the survey weights aligned the study sample with the U.S. adult population based on the U.S. Current Population Survey. [[19]]’ https://www.thelancet.com/journals/lanam/article/PIIS2667-193X%2821%2900087-9/fulltext Could also add the decimal place: 64.3

R1 Response 7

Thank you for these suggestions. We replaced “64% completed” with “64.3% completion rate”. We also added the following clarifying sentence: “We created and applied post-stratification weights to align the study sample with the U.S. adult population according to the U.S. Current Population Survey [15].” 

R1 Comment 8

Page 3, line 73: We excluded participants who did not respond to questions about suicidal ideation in NHANES and participants who did not respond to any analysis variables in CLIMB data.’ – I presume the results section will mention how many people were excluded?

R1 Response 8

Thank you for this comment. We included the following sentence in the results section that describes the total number of survey respondents as well as the number of participants included in the analytic sample: “A total of 1,415 (96.3%) of 1,470 CLIMB participants responded to all questions relevant to the analysis and 5,085 (86.8%) of 5,856 NHANES participants responded to suicidal ideation questions and were included in the samples.”

R1 Comment 9

Page 4, line 77: ‘We evaluated three COVID-19 stressors reflecting economic precarity and social isolation, each measured as binary variables reported in response to a question, “Have any of the following affected your life as a result of the coronavirus or COVID-19 outbreak?” First, we measured job loss based on checking “losing a job.” Second, we measured difficulty paying rent based on checking “having difficulty paying rent.” Third, we measured social isolation as checking “feeling alone.”’

– I think the authors’ findings are really interesting and important as it is nice to see a focus on inequity with these outcomes. So, could the authors clarify that these were the only available stressors for economic precarity and social isolation, and if there were reasons for not looking at other aspects (e.g., outside the scope of the paper)?

Also, it may be odd to say ‘we measured’ if you did not select the measures or collect the data. You could say “The three checkboxes we used asked about ‘losing a job,’ ‘having difficulty paying rent,’ and “feeling alone.”

R1 Response 9

Thank you for this comment. Although the survey measured additional stressors including travel restrictions and not being able to get food due to shortages, we chose to focus on economic precarity and loneliness in particular. We inserted the following sentence in place of describing the exposure variables as ones we “measured”: “Our exposure variables were based on whether participants checked boxes for “losing a job,” “having difficulty paying rent,” and “feeling alone”.

R1 Comment 10 

Page 4, line 85: ‘We measured suicidal ideation’ – could say that ‘both surveys assessed suicidal ideation.’

R1 Response 10

Thank you for this suggestion. The sentence now begins with the following: “Both surveys assessed suicidal ideation…”. 

R1 Comment 11

Page 4, line 89: ‘Prior research indicates responses to this question were correlated with future suicide attempts and deaths [7-9].’ – if they are future suicide attempts and deaths, do you need to say ‘predict’ instead of ‘were correlated with’, and say how well they predict, as any one risk factor by itself is likely a poor predictor of attempts and deaths – see https://pubmed.ncbi.nlm.nih.gov/27841450/

R1 Response 11

Thank you for this comment. As you highlight, any one risk factor is a poor predictor of suicide attempts and deaths. Because of this, we chose to use language that describes an association to avoid implying that a non-RCT is causal.

R1 Comment 12

Page 4, line 96: It is great the authors have opted with PRs and Poisson rather than ORs and log reg. Is it easy for the authors to supply their code for the analysis in case other researchers may wish to repeat their study with similar data?

R1 Response 12

Thank you so much for this suggestion. We are happy to supply our code for the analysis.

R1 Comment 13

Page 5, line 106: ‘Overall, suicidal ideation increased more than fourfold, from 3.4% in the 2017-2018 NHANES to 16.3% in the 2020 CLIMB survey.’ – As it’s closer to fivefold the authors could say ‘4.79’ or ‘4.8,’ perhaps one decimal place is best.

R1 Response 13

Thank you for this suggestion. Based on feedback from the editor, we replaced this sentence with the following: “The prevalence of suicidal ideation was 3.4% in the 2017-2018 NHANES sample and 16.3% in the 2020 CLIMB survey.”

R1 Comment 14

Page 6, line 116 and page 8, line 147. Can the authors please have columns reporting precise rather than thresholded p-values? This will avoid Type I and Type II errors

(https://www.ncbi.nlm.nih.gov/pmc/articles/PMC2850991/), align with the ASA Statement on p-values (https://www.tandfonline.com/doi/full/10.1080/00031305.2016.1154108) and guidelines from other influential papers on p-values (https://www.ncbi.nlm.nih.gov/pmc/articles/PMC4877414/).

R1 Response 14

Thank you for this comment. We divided table 2 into two tables (tables 2 and 3) and added columns to each of the tables that report precise p-values.

R1 Comment 15

Page 9, line 156: ‘more than fourfold’ – 4.8 times higher? Or nearly fivefold.

R1 Response 15

Thank you for this suggestion. We replaced “more than fourfold increase” with “4.8 times higher”.

R1 Comment 16

Page 9, line 164: ‘Center for Disease Control and Prevention’ – ‘Centers’

R1 Response 16

Thank you for catching this error. We have made the correction.

R1 Comment 17

Page 9, line 166: ‘may play help’ – typo?

R1 Response 17

Thank you for pointing this out. We corrected the typo and it now reads “may help”.

R1 Comment 18

Page 9, line 172: ‘People who reported feeling lonely were nearly twice as likely to report suicidal ideation’ – 1.9 times as likely? Also, there is some inconsistency in terminology here. The question AmeriSpeak asked was about feeling alone, which you’ve termed social isolation, but it’s referred to as ‘feeling lonely’ in the discussion. Maybe for consistency and brevity you can say ‘feeling alone’ throughout, as it depends on how people interpreted this question (were they actually socially isolated, or perceiving being socially isolated and feeling alone and lonely).

R1 Response 18

Thank you for this comment. We replaced “nearly twice” with “1.9 times.” We also replaced “feeling lonely” with “feeling alone”. Throughout the manuscript, we adjusted the language we used to say “feeling alone” or “loneliness” instead of “social isolation”.

R1 Comment 19

Page 10, line 167: ‘While job loss was not associated with suicidal ideation during the CLIMB study period of late March and early April,’ – do you need to suffix with ‘in the adjusted analysis’?

R1 Response 19

Thank you for this suggestion. We added the following phrase: “in the adjusted analysis”. 

R1 Comment 20

Page 10, line 183: ‘particularly of firearms [15], is associated with reductions in suicide.’

- A lot of the means restriction interventions were recently summarised in: Ishimo, M. C., Sampasa-Kanyinga, H., Olibris, B., Chawla, M., Berfeld, N., Prince, S. A., ... & Lang, J. J. (2021). Universal interventions for suicide prevention in high-income Organisation for Economic Co-operation and Development (OECD) member countries: a systematic review. Injury prevention, 27(2), 184-193. I suggest citing it.

For firearms specifically, I think the latest systematic review is this, but I suggest the authors citation search it: https://academic.oup.com/epirev/article/38/1/140/2754868?login=true Also better to cite though than the currently cited study.

You might also need to say here that firearms is the predominant suicide method in the US (as it might be hanging elsewhere) so that international readers know why firearms warrant mentioning here.

R1 Response 20

Thank you for these suggestions. We cited both reviews you recommended in place of the previously cited studies. We also added clarification that firearms is the predominant suicide method in the US.

R1 Comment 21

Page 10, line 188 ‘Limitations include that suicidal ideation was based on self-report’ – yes, any self-reported measure has limitations but how else do we measure ideation? If there is not viable alternative way to measure ideation, I suggest the authors omit this as it’s not something that can be improved upon and it’s a universal limitation of studies on ideation.

R1 Response 21

Thank you for this suggestion. We removed this limitation as you recommended.

R1 Comment 22

Page 10, line 188 ‘that the characteristics of participants in CLIMB and NHANES differed.’ – But both were nationally representative surveys? Additionally, you should probably note here that suicidal ideation was still higher in 2020 (I don’t mean ‘statistical significance,’ just higher) across all strata, compared to 2017 to 2018.

R1 Response 22

Thank you for this comment. We added a clarifying phrase to the limitations that both surveys were nationally representative. We also added the following sentence to describe how the characteristics of participants differed between the two surveys: “The NHANES sample was younger and less likely to be low-income.”

R1 Comment 23

Page 10, line 189 ‘Those who responded to surveys may have differed from those who did not, particularly if stressors affected survey participation.’ – I’m not a survey expert, but as I understand, post-stratification weighting should cancel this out by adjusting for non-response?

R1 Response 23

Thank you for this comment. We removed this sentence from the manuscript.

R1 Comment 24

Page 10, lines 188-193 – It would perhaps be good to include something about the limitations or exposure measurement if you think there are some. 1-item measures are sometimes criticised. ‘having difficulty paying rent’ is fairly straightforward, although it does not account for people’s income levels. ‘Losing a job’ could mean different things to different people – what about a contractor who has lots of small jobs and loses one job. ‘Feeling alone’ seems a little open to interpretation. These are only minor limitations, but may just warrant mentioning.

R1 Response 24

Thank you for this suggestion. We added the following sentence to the discussion section: “We were further limited by the questions asked in the surveys as indicators of economic precarity and loneliness and the possibility that participants interpreted these questions in different ways.”

R1 Comment 25

Page 10, line 186: Those facing difficulty paying rent and loneliness may be at particular risk of suicide.’ – maybe ‘feeling alone’ instead of ‘loneliness’ for consistency?

R1 Response 25

Thank you for this suggestion. We changed “loneliness” to “feeling alone”.

R1 Comment 26

Page 14, Fig 1: I’m not sure the data labels add to the figure as they are already in the tables, so they could be removed. As they are above the upper limit, they might be misinterpreted as the upper limit.

R1 Response 26

Thank you for this comment. We removed the data labels from the figure.

R1 Comment 27

Page 14, Fig 1: Red is sometimes considered a colour to avoid in the suicide sector due to the links to blood. The authors could instead choose from these colorblind-friendly colors: https://jfly.uni-koeln.de/color/#pallet

R1 Response 27

Thank you for this suggestion and for sharing the colorblind-friendly resource. We changed the color scheme of the figure.

R1 Comment 28

As this is an observational study, per PLOS One requirements the authors need to report according to, populate and submit the STROBE Statement. If the authors consider that NHANES and AmeriSpeak are routinely collected health data, the RECORD Statement (considered supplementary to STROBE and submitted alongside it) can be populated and submitted as well.

R1 Response 28

Thank you for this comment. We completed and submitted the STROBE Statement with the resubmission. 

R1 Comment 29

Overall, I think this is a really good manuscript. As the literature on suicide is moving so quickly during COVID-19 I’d encourage the authors to check the literature for new studies and reviews just prior to submission if invited to resubmit, be as outcome-specific as possible (ideation, not attempts and deaths), and emphasise slightly more the novelty of this paper (have many other papers done this?) focusing on inequities and people disproportionately bearing the impacts of COVID-19.

R1 Response 29

Thank you so much for your helpful and thoughtful feedback. We revised the introduction to focus more specifically on suicidal ideation with references to recent literature in the context of the COVID-19 pandemic. We highlighted how our paper is unique in that it explores a relationship between economic precarity and suicidality and complements other work related to suicidal ideation during the pandemic. We also added the following sentence to the introduction to describe how historical and modern-day policies shaped inequities in COVID-19 and economic outcomes: “Black, Hispanic, and Native American people are made especially vulnerable to these economic shocks as a result of centuries of structural racism that shape inequities in wealth.” 

Reviewer #2: This is an interesting study on the increase in the prevalence of suicidal ideation during the COVID-19 pandemic.

However, I do not recommend this article for publication in its current form as several limitations have to be highlighted:

R2 Comment 1

"The CLIMB and NHANES samples are comparable in that both are nationally representative": in my opinion, this statement is one of the major limitations of the study and should be more clearly analyzed and discussed. The Table 2 should try to depict more precisely the comparability of the two samples. Notably, the percentage of people with high school graduate or less is higher in the NHANES compared to the CLIMBS.

R2 Response 1

Thank you for your comment. We appreciate the encouragement to more formally compare the samples. We separated Table 2 into two tables to fully describe the sample and we calculated p-values for differences in proportions. We described differences between the samples: “The NHANES sample was younger, less likely to be married, more likely to have high school or less education, and less likely to be low-income relative to the CLIMB sample.”

R2 Comment 2

The prevalence of "feeling alone", "difficulty paying a rent" or "lost job" were not assessed in the NHANES study so that it is not possible to know if the prevalence of these social conditions were similar or not in the two samples. Moreover, economic precarity and loneliness are already identified as robust risk factors for suicidal ideation and behaviors so that this result does not add much value to the existing evidence regarding suicide prevention.

R2 Response 2

Thank you for your comment. We thought it was important to evaluate economic hardship and loneliness as risk factors during the COVID-19 pandemic, given that they rose in prevalence. We also added discussion of the limitation in measurement in NHANES prior to the pandemic to the discussion section: “We were further limited by the questions asked in the surveys as indicators of economic precarity and loneliness and the possibility that participants interpreted these questions in different ways. These questions were not included in the NHANES survey, so it was not possible to compare these exposures before and after the pandemic.”

R2 Comment 3

It is not clear if the prevalence ratios were measured with the data issued from the two samples or only for the CLIMB study? As some variables were measured in the two samples and other only in the CLIMB sample, the statistic analyses should be more precisely described and explained.

R2 Response 3

Thank you for this comment. We added clarification to the analysis subsection of the Methods to describe that the prevalence ratios were measured with data from the CLIMB study only.

R2 Comment 4

Figure 1: the p should be given for a better understanding of the results depicted in this figure. Are the observed differences statistically significant?

R2 Response 4

Thank you for this suggestion. We added p-values to the figure to demonstrate that the observed differences are statistically significant. 

R2 Comment 5

The discussion does not fit with the results of the study. For example, the authors discuss the role of means restrictions to prevent suicide, while their study is focused on suicidal ideation with no assessment of access to suicide means.

R2 Response 5

Thank you for this comment. We appreciate your point and agree that the focus of this manuscript is on suicidal ideation. However, knowing that suicidality is related to suicide, we think it is important to additionally speak to strategies for suicide prevention. We considered your thoughts and the comments from Reviewer 1 and we added the following sentences to the discussion section: “Suicide by firearm is the predominant means of suicide death in the United States. As such, and in light of previous evidence that links suicidality to suicide death, policies or programs to reduce household firearm ownership could play an important role in suicide prevention in the COVID-19 context of elevated stressors.”

R2 Comment 6

"We found that people living in low-income households and young people are particularly at risk of mental distress during the COVID-19 pandemic": this statement is not supported by the multivariate analysis showing no differences in reported suicidal ideation according to the age of participants

R2 Response 6

Thank you for this comment. We removed the phrase “and young people” from this sentence.

---

## [Decision Letter · Decision Letter 1]

28 Jun 2022

PONE-D-21-30553R1Economic precarity, loneliness, and suicidal ideation during the COVID-19 pandemicPLOS ONE

Dear Dr. Raifman,

Thank you for submitting your manuscript to PLOS ONE. After careful consideration, we feel that it has merit but does not fully meet PLOS ONE’s publication criteria as it currently stands. Therefore, we invite you to submit a revised version of the manuscript that addresses the points raised during the review process.

There are a few minor issues still pending and raised by reviewer #1, which need some attention.

We look forward to receiving your revised manuscript.

Kind regards,

Pedro Vieira da Silva Magalhaes, M.D., Ph.D.

Academic Editor

PLOS ONE

Journal Requirements:

Reviewers' comments:

Reviewer's Responses to Questions

**Comments to the Author**

1. If the authors have adequately addressed your comments raised in a previous round of review and you feel that this manuscript is now acceptable for publication, you may indicate that here to bypass the “Comments to the Author” section, enter your conflict of interest statement in the “Confidential to Editor” section, and submit your "Accept" recommendation.

Reviewer #1: (No Response)

Reviewer #2: All comments have been addressed

2. Is the manuscript technically sound, and do the data support the conclusions?

Reviewer #1: Yes

Reviewer #2: Yes

3. Has the statistical analysis been performed appropriately and rigorously? 

Reviewer #1: Yes

Reviewer #2: Yes

4. Have the authors made all data underlying the findings in their manuscript fully available?

Reviewer #1: No

Reviewer #2: Yes

5. Is the manuscript presented in an intelligible fashion and written in standard English?

Reviewer #1: Yes

Reviewer #2: Yes

6. Review Comments to the Author

Reviewer #1: Thank you to the authors for making these revisions, and I accept their rebuttals on some points. I have no further comments on how they addressed the revisions. Some minor points:

Page 2, line 33: after ‘robust variance’, just add ‘to generate unadjusted and adjusted prevalence ratios (PR and aPR).’ – only aPR is mentioned in the abstract, so can omit ‘PR and’ if space is tight.

Page 2, line 38: for balance eporting null and non-null findings and completeness in the abstract, you could add to the end of this sentence: ‘but job loss was not (aPR: 0.9, 95% CI: 0.6 to 1.2)’

Page 2, lines 39-40: Should move this sentence to line 35, after ‘households.’ as it’s the total sample.

Page 13, lines 185-189: Job loss may also be a crude marker as someone might obtain another job relatively quickly while others may not, and I think people more disadvantaged would find it more difficult to get another job? Also, as you have shown, there is a lower prevalence of suicidal ideation in those with more savings, so job loss will affect people differently (some people may not have liked their job and liked their new job better, too, if they found one). The number losing their job in CLIMB 2020 is perhaps too small to stratify by those variables, so perhaps that’s a subsequent study you can suggest to understand this more. Initial relationships (or absence - near parity in the PR) might look different down the track when you can stratify this by other important variables.

Page 13, lines 191-194: ‘Feeling alone’ is in the literature and my experience, more frequently associated with the suicides of elderly people, and we know they likely use technology less. This is perhaps an area for future study – a stratified analysis by age group to see if these recommendations are broad-based or apply to specific groups.

Page 15, lines 220-221: Could add ‘there is a need for further research to determine if job loss affects suicidal ideation differently in different population subgroups’, or something to that effect.

Thank you for the opportunity to review this manuscript again.

Reviewer #2: (No Response)

7. PLOS authors have the option to publish the peer review history of their article (what does this mean?). If published, this will include your full peer review and any attached files.

Reviewer #1: **Yes: **Stuart Leske

Reviewer #2: **Yes: **Edouard Leaune

---

## [Author Response · Author response to Decision Letter 1]

26 Sep 2022

Thank you for the opportunity to revise and respond to the reviewers’ comments on our manuscript, entitled “Economic precarity, social isolation, and suicidal ideation during the COVID-19 pandemic.” We appreciate the comments and the editor’s and reviewers’ thoughtful feedback that has improved the quality of this manuscript. We have provided detailed responses below and tracked changes to the manuscript. 

Reviewer #1: Thank you to the authors for making these revisions, and I accept their rebuttals on some points. I have no further comments on how they addressed the revisions. Some minor points:

Thank you again for your thoughtful reviews and for your additional suggestions.

Page 2, line 33: after ‘robust variance’, just add ‘to generate unadjusted and adjusted prevalence ratios (PR and aPR).’ – only aPR is mentioned in the abstract, so can omit ‘PR and’ if space is tight.

Thank you for the suggestion. We made this change.

Page 2, line 38: for balance eporting null and non-null findings and completeness in the abstract, you could add to the end of this sentence: ‘but job loss was not (aPR: 0.9, 95% CI: 0.6 to 1.2)’

Thank you for the suggestion. We made this change.

Page 2, lines 39-40: Should move this sentence to line 35, after ‘households.’ as it’s the total sample.

Thank you for the suggestion. We made this change.

Page 13, lines 185-189: Job loss may also be a crude marker as someone might obtain another job relatively quickly while others may not, and I think people more disadvantaged would find it more difficult to get another job? Also, as you have shown, there is a lower prevalence of suicidal ideation in those with more savings, so job loss will affect people differently (some people may not have liked their job and liked their new job better, too, if they found one). The number losing their job in CLIMB 2020 is perhaps too small to stratify by those variables, so perhaps that’s a subsequent study you can suggest to understand this more. Initial relationships (or absence - near parity in the PR) might look different down the track when you can stratify this by other important variables.

Thank you for this suggestion. We added a sentence about this caveat and the suggestion for further study.

“It is also possible that the suicidality impacts of job loss differed by wealth and whether people who lost work faced imminent economic hardship, and further studies among subgroups are needed.”

Page 13, lines 191-194: ‘Feeling alone’ is in the literature and my experience, more frequently associated with the suicides of elderly people, and we know they likely use technology less. This is perhaps an area for future study – a stratified analysis by age group to see if these recommendations are broad-based or apply to specific groups.

Thank you for this suggestion. We added the following sentence to the discussion of feeling alone.

“There is a need for further research on loneliness among subgroups such as older populations.”

Page 15, lines 220-221: Could add ‘there is a need for further research to determine if job loss affects suicidal ideation differently in different population subgroups’, or something to that effect.

Thank you for this suggestion. We added the following sentence to the discussion section.

“There is a need for further research on exposures and suicidality over time and among subgroups, especially those affected by structural racism and inequities.”

---

## [Editor Report · Decision Letter 2]

27 Sep 2022

Economic precarity, loneliness, and suicidal ideation during the COVID-19 pandemic

PONE-D-21-30553R2

Dear Dr. Raifman,

We’re pleased to inform you that your manuscript has been judged scientifically suitable for publication and will be formally accepted for publication once it meets all outstanding technical requirements.

Kind regards,

Pedro Vieira da Silva Magalhaes, M.D., Ph.D.

Academic Editor

PLOS ONE
---

## [Editor Report · Acceptance letter]

8 Nov 2022

PONE-D-21-30553R2 

Economic precarity, loneliness, and suicidal ideation during the COVID-19 pandemic 

Dear Dr. Raifman:

I'm pleased to inform you that your manuscript has been deemed suitable for publication in PLOS ONE. Congratulations! Your manuscript is now with our production department. 

Kind regards, 

on behalf of

Professor Pedro Vieira da Silva Magalhaes 

Academic Editor

PLOS ONE